# Early stage prion assembly involves two subpopulations with different quaternary structures and a secondary templating pathway

Angélique Igel-Egalon [1,6], Florent Laferrière [1,5,6], Mohammed Moudjou [1,6], Jan Bohl[1,2], Mathieu Mezache[1,3], Tina Knäpple[1], Laetitia Herzog[1], Fabienne Reine[1], Christelle Jas-Duval[1,4], Marie Doumic[3], Human Rezaei[1,7]* & Vincent Béringue [1,7]*

The dynamics of aggregation and structural diversification of misfolded, host-encoded proteins in neurodegenerative diseases are poorly understood. In many of these disorders, including Alzheimer's, Parkinson's and prion diseases, the misfolded proteins are self-organized into conformationally distinct assemblies or strains. The existence of intrastrain structural heterogeneity is increasingly recognized. However, the underlying processes of emergence and coevolution of structurally distinct assemblies are not mechanistically understood. Here, we show that early prion replication generates two subsets of structurally different assemblies by two sequential processes of formation, regardless of the strain considered. The first process corresponds to a quaternary structural convergence, by reducing the parental strain polydispersity to generate small oligomers. The second process transforms these oligomers into larger ones, by a secondary autocatalytic templating pathway requiring the prion protein. This pathway provides mechanistic insights into prion structural diversification, a key determinant for prion adaptation and toxicity.

[1] VIM, INRA, Université Paris-Saclay, 78350 Jouy-en-Josas, France. [2] LCP, CNRS, Université Paris Sud, 91400 Orsay, France. [3] INRIA, MAMBA, Université Paris VI, 75005 Paris, France. [4] Pathogenesis and Control of Chronic Infections, EFS, INSERM, University of Montpellier, 34000 Montpellier, France. [5] Present address: Institute of Neurodegenerative Diseases, CNRS UMR5293, University of Bordeaux, Bordeaux, France. [6] These authors contributed equally: Angélique Igel-Egalon, Florent Laferrière, Mohammed Moudjou. [7] These authors jointly supervised this work: Human Rezaei and Vincent Béringue. *email: human.rezaei@inra.fr; vincent.beringue@inra.fr

In terms of pathogenic mechanisms, the prion paradigm unifies a number of neurodegenerative disorders that are caused by protein misfolding and aggregation[1–4]. These disorders include Alzheimer's, Parkinson's, Huntington's, and prion diseases. In principle, host-encoded monomeric proteins are converted into misfolded and aggregated assemblies, which serve as templates for further autocatalytic conversion. In prion diseases, the prion protein PrP$^C$ is converted into a misfolded, β-sheet-rich conformer termed PrP$^{Sc}$[5]. In susceptible hosts, PrP$^{Sc}$ assemblies form stable, structurally distinct PrP$^{Sc}$ conformers termed prion strains[6–9], which encode stereotypical biological phenotypes[10–13]. The strain-specific differences can be observed at the secondary and tertiary structural level in terms of local structural variation but also at the quaternary level with strain-specific size distribution signature[11,14,15]. A large body of evidence supports the view for structural diversity within specific prion populations and strains: first, prion substrains can be preferentially selected during prion transmission[16–19] with a species/transmission barrier; second, size- or density-fractionation studies support the existence of a heterogeneous spectrum of PrP$^{Sc}$ assemblies with distinct tertiary/quaternary structures[14,15,20–25] and biological activity (templating activity and infectivity)[14,15,20], and third, kinetic studies of prion pathogenesis suggest that the formation of neurotoxic PrP$^{Sc}$ species[26] occurs at the late stage of prion infection when replicative PrP$^{Sc}$ assemblies are formed at earlier stages[27,28]. The prion replication process thus intrinsically allows the structural diversification of PrP$^{Sc}$ assemblies.

While the kinetic aspects of prion replication "as a whole" have been comprehensively described by measuring infectivity or PrP$^{Sc}$ levels in the brain (e.g., references[29,30]), PrP$^{Sc}$ structural diversification processes remain undescribed and are not mechanistically supported by prion paradigm frameworks. The autocatalytic conversion model proposed by Griffith in 1967[31], the nucleated-polymerization model described by Lansbury and Caughey in 1995[32] and other derived models (e.g.[33]) merely assume the existence of structurally homogenous assemblies that have absolutely identical propensity to replicate throughout disease progression. These mechanisms intrinsically reduce PrP$^{Sc}$ heterogeneity due to the best replicator selection process[34,35,36]. A recent high-resolution structural analysis of the N-terminal domain of the yeast prion SuP35 suggests that conformational fluctuations in natively disordered monomeric Sup35 are responsible for the stochastic, structural diversification of Sup35 aggregates[37]. While this idea may be extrapolated to mammalian prion PrP to explain intrastrain structural diversification and strain mutation[8], it does not explain the copropagation of structurally distinct PrP$^{Sc}$ subassemblies in the same environment[16,38].

To examine the molecular mechanisms of PrP$^{Sc}$ replication and structural diversification, we explore, by sedimentation velocity (SV)-based methods, the quaternary structure evolution of PrP$^{Sc}$ assemblies during the early stage of prion conversion in vivo and in a cell-free system by protein misfolding cyclic amplification[39] (PMCA). By using several prion strains as templates, we demonstrate that early prion replication invariably generates two subsets of assemblies, termed A$_i$ and B$_i$, which differ in proportion, size, the architecture of their elementary bricks and structure. Analyzing their kinetics of formation during PMCA together with kinetic data assimilation reveals the existence of two sequential processes of formation. The first process corresponds to a quaternary structural convergence, as it tends to reduce the parental quaternary structure polydispersity to generate mostly small-sized assemblies, namely A$_i$. The second process transforms the A$_i$ into structurally different assemblies, namely B$_i$, according to a secondary auto-catalytic pathway requiring PrP$^C$, whereby B$_i$ facilitates its own formation.

## Results

### Small PrP$^{Sc}$ oligomers are formed at early replication stage.

The early phases of prion replication are commonly thought to consist of an elongation process[40], with the PrP$^{Sc}$ template serving as a base. We studied the size distribution of proteinase K (PK)-resistant PrP$^{Sc}$ (PrP$^{res}$) assemblies at the early step of prion replication in the brain by SV in an iodixanol gradient using a previously published methodology[15,20,41]. The PrP$^{res}$ size distribution at the disease end-stage served as control. Three different host-PrP/strain combinations were studied: the 127S cloned scrapie prion strain in ovine PrP tg338 transgenic mice[29], the 139A cloned mouse scrapie prion strain in murine PrP tga20 mice[42] and the vCJD cloned human prion strain in human PrP tg650 mice[43,44]. As shown in Fig. 1a–c (Supplementary Fig. 1a–c for the corresponding immunoblots), small oligomers sedimenting between fractions 1 and 4 were preferentially detected at the early stage of pathogenesis, regardless of the strain considered. A second population of oligomers with a larger size distribution and peaking in fractions 8–10 and 18 was observed for 127S prions. The contribution of incompletely digested PrP$^C$ or remnant PrP$^{Sc}$ inoculum to the PrP signal detected in the top fractions was discarded. No PrP$^C$ signal was detected after PK treatment of uninfected tg338 brain (Supplementary Fig. 2a–b). No PrP$^{res}$ was detected in the brain of vCJD inoculated PrP knock-out mice (PrP$^{-/-}$) analysed for PrP$^{res}$ content at early time points (Supplementary Fig. 3). At the disease end stage and for the three strains, larger assemblies were observed (Fig. 1a–c, Supplementary Fig. 1a–c). These observations suggest that the replication process follows two phases that are common to the three strains: (i) a first phase generating mainly small oligomers equivalent in size; (ii) a second phase of quaternary structural diversification during the disease evolution.

We next determined whether these phases can be reproduced by an in vitro bona fide amplification method. We used a high-throughput variant of PMCA (termed mb-PMCA[41,45,46]). mb-PMCA generates in one unique round of 48 h as much infectivity as in the brain at the terminal stage of the disease, with high reproducibility in terms of limiting dilution and amplification yield[45,46]. When the size distribution of the amplified products was analysed by SV, two discrete distributions were observed for the three strains, a major set of small PrP$^{res}$ assemblies sedimenting between fractions 1 and 3 (named peak P$_1$) and a minor set of larger assemblies with a well-defined Gaussian distribution centered on fraction 15 (named peak P$_2$) (Fig. 1d; Supplementary Fig. 1d). The relative proportions of P$_1$ and P$_2$ varied among the three strains; P$_2$ was barely detected in the 139A amplicons. When the mb-PMCA reaction was seeded with healthy tg338 brain homogenate, there was no evidence of spontaneous formation of P$_1$ and P$_2$ PrP$^{res}$ in the amplified products (Supplementary Fig. 2c–d). These data indicate that mb-PMCA generates two populations of PrP$^{Sc}$ assemblies that differ according to their quaternary structures, with a predominance of small assemblies.

The bimodal (i.e., generation of two peaks) and discrete behavior of the size distribution as well as the formation of predominantly small assemblies in P$_1$ may originate from the mb-PMCA conditions (i.e. shearing forces during the sonication step[47–49]) rather than from the replication process itself. To discriminate between these two possibilities, undiluted 127S seeds (i.e., 20% brain homogenate) were incubated and sonicated in identical mb-PMCA conditions, but without the PrP$^C$ substrate (i.e., 1:1 dilution in PrP$^{0/0}$ brain lysate). The samples were then SV-fractionated and analysed for PrP$^{res}$ content by western blot. For comparison, the same brain was diluted in the PMCA buffer immediately before the mb-PMCA reaction, as we reported previously that a simple dilution affects the size distribution of

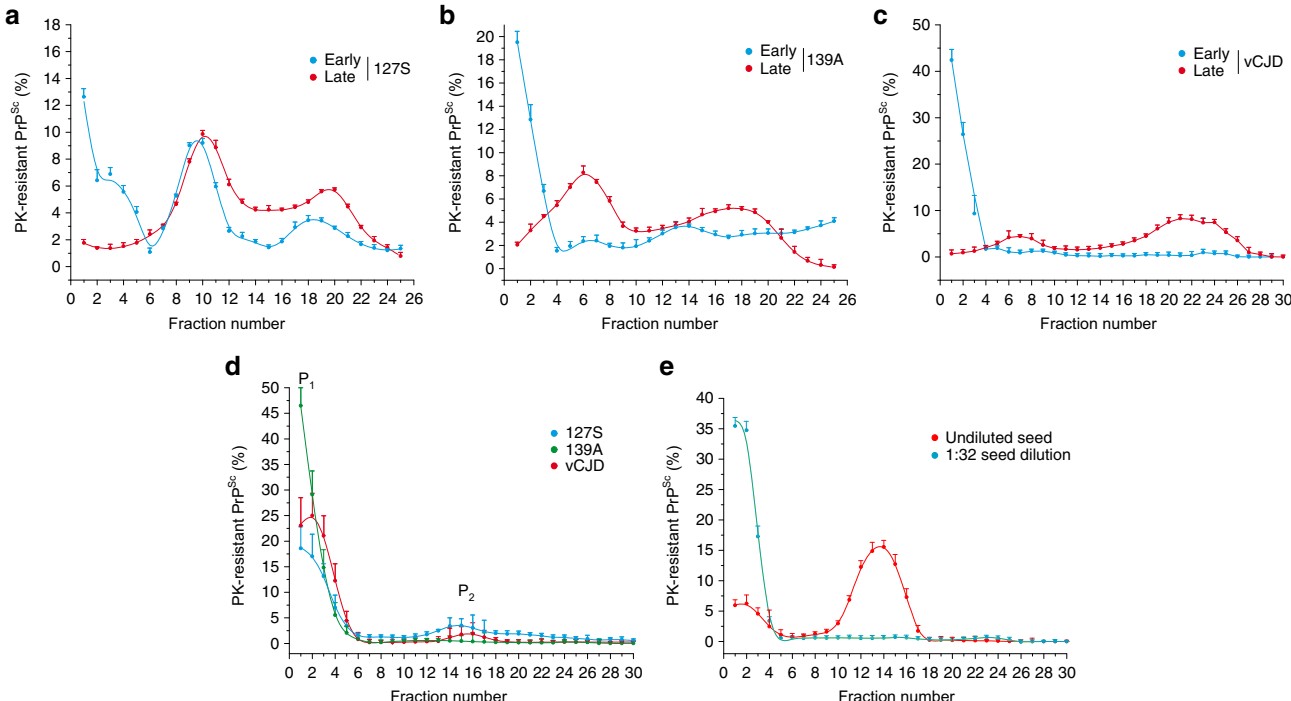

**Fig. 1** Size distribution of PrP$^{Sc}$ assemblies from different prion strains at the early and late stages of pathogenesis in vivo and after the PMCA reaction. The size distribution of proteinase K (PK)-resistant PrP$^{Sc}$ assemblies present in the brain in vivo (**a**–**c**) and in PMCA products (**d**, **e**) was examined by sedimentation velocity (SV). **a**–**c** For the in vivo sedimentograms, brains from ovine (tg338), murine (tga20) and human (tg650) transgenic mice inoculated with 127S scrapie prions (**a**), 139A mouse prions (**b**), and vCJD human prions (**c**) were collected (in triplicate) at the early stage (15 days postinfection (127S), 11 days postinfection (139A) and 120 days postinfection (vCJD), blue curves) and at the end stage of the disease (60 days postinfection (127S), 55 days postinfection (139A), 495 days postinfection (vCJD), red curves). The brains were solubilized and SV-fractionated. The collected fractions (numbered from top to bottom) were analysed for PK-resistant PrP$^{Sc}$ content by immunoblotting (mean ± SEM values obtained from $n = 3$ independent fractionations, representative immunoblots at early and late stage shown in Supplementary Fig. 1). **d**, **e** For the sedimentograms from the PMCA products with PrP$^{C}$ substrate (**d**), the same prion strains were subjected to a single round of mb-PMCA by using $10^{-5}$ (139A) or $10^{-6}$ (vCJD, 127S) diluted brain homogenates as seed for the reaction. Thirty minutes after the last sonication, the amplified products were solubilized and SV-fractionated. The mean ± SEM levels of PK-resistant PrP$^{Sc}$ per fraction were obtained from the immunoblot analysis of $n = 4$ independent fractionations of PMCA reactions. The peaks containing PrP$^{Sc}$ assemblies sedimenting in the top and middle fractions were termed $P_1$ and $P_2$, respectively. For the sedimentograms from the PMCA products without PrP$^{C}$ substrate (**e**), undiluted 127S-infected tg338 brain (20% w/v, red curve) or a 1:32 dilution in PMCA buffer (blue curve) was used as seed, mixed with equal volume of brain homogenate from PrP$^{0/0}$ mice as substrate and subjected to a single round of mb-PMCA before SV fractionation (mean ± SEM levels from $n = 3$ independent fractionations, representative immunoblots shown in Supplementary Fig. 1)

PrP$^{Sc}$ assemblies, by displacing the equilibrium between PrP$^{Sc}$ assemblies and their suPrP elementary subunit $(\text{PrP}_i^{Sc} \rightleftharpoons \text{PrP}_{i-1}^{Sc} + \text{suPrP})$[41,50]. While the dilution of 127S seed indeed drastically affected the size distribution of 127 S PrP$^{Sc}$ assemblies, sonication of concentrated 127S seeds in the PrP$^{0/0}$ substrate revealed mostly the presence of large-sized assemblies (Fig. 1e, Supplementary Fig. 1), peaking in fraction 12–16, as for 127S fractionated brain material solubilized at 37 °C[20]. The absence of sonication effect on PrP$^{Sc}$ assemblies size distribution rules out a fragmentation process during the mb-PMCA being at the origin of the formation of small-size assemblies.

Altogether, these observations suggest that (i) in vivo, the early phase of replication generates mainly small-sized assemblies, which diversify with respect to quaternary structure during the disease pathogenesis, (ii) Similar to in vivo replication, the mb-PMCA amplification conditions generate two sets of PrP assemblies that differ in their quaternary structures. The formation of these two groups of assemblies is common to the 127S, 139A, and vCJD strains.

**Bimodal and autocatalytic evolution of PrP$^{res}$ from $P_1$ to $P_2$.** We next asked whether $P_2$ formation resulted from a simple condensation of $P_1$ peak assemblies (Oswald ripening[51] or

coagulation[52,53] process) or from an alternative templating pathway. We first examined the influence of the amplification rate on the formation of these two species by varying the concentration of the seed used as template for the mb-PMCA reaction. We generated mb-PMCA products seeded with $10^{-3}$–$10^{-10}$ dilutions of 127S brain homogenate. The amounts of PrP$^{res}$ amongst the amplified products were similar, whatever the seed dilution (Supplementary Fig. 4), as previously observed[46]. The SV-sedimentograms of the mb-PMCA products are shown in Fig. 2a. The relative amounts of assemblies in $P_1$ decreased as the amounts of those from $P_2$ increased as a function of the seed concentration. The variation in the $P_1$ and $P_2$ peak maximum as a function of the logarithm of the dilution factor revealed a quasi-linear decrease in $P_1$ when the $P_2$ peak maximum followed a sigmoidal increase (Fig. 2b). Such non-linear correlation in $P_1$ and $P_2$ peak variation indicates that (i) $P_2$ peak formation does not result from the simple condensation of assemblies present in $P_1$, (ii) the formation of PrP$^{res}$ assemblies in $P_2$ follows a seed concentration-dependent cooperative process. This strongly suggests that the assemblies forming the $P_1$ and $P_2$ peaks result from distinct polymerization pathways and should therefore be structurally distinct.

To further explore the entanglement between the assemblies forming $P_1$ and $P_2$, we set the mb-PMCA regime to favor the

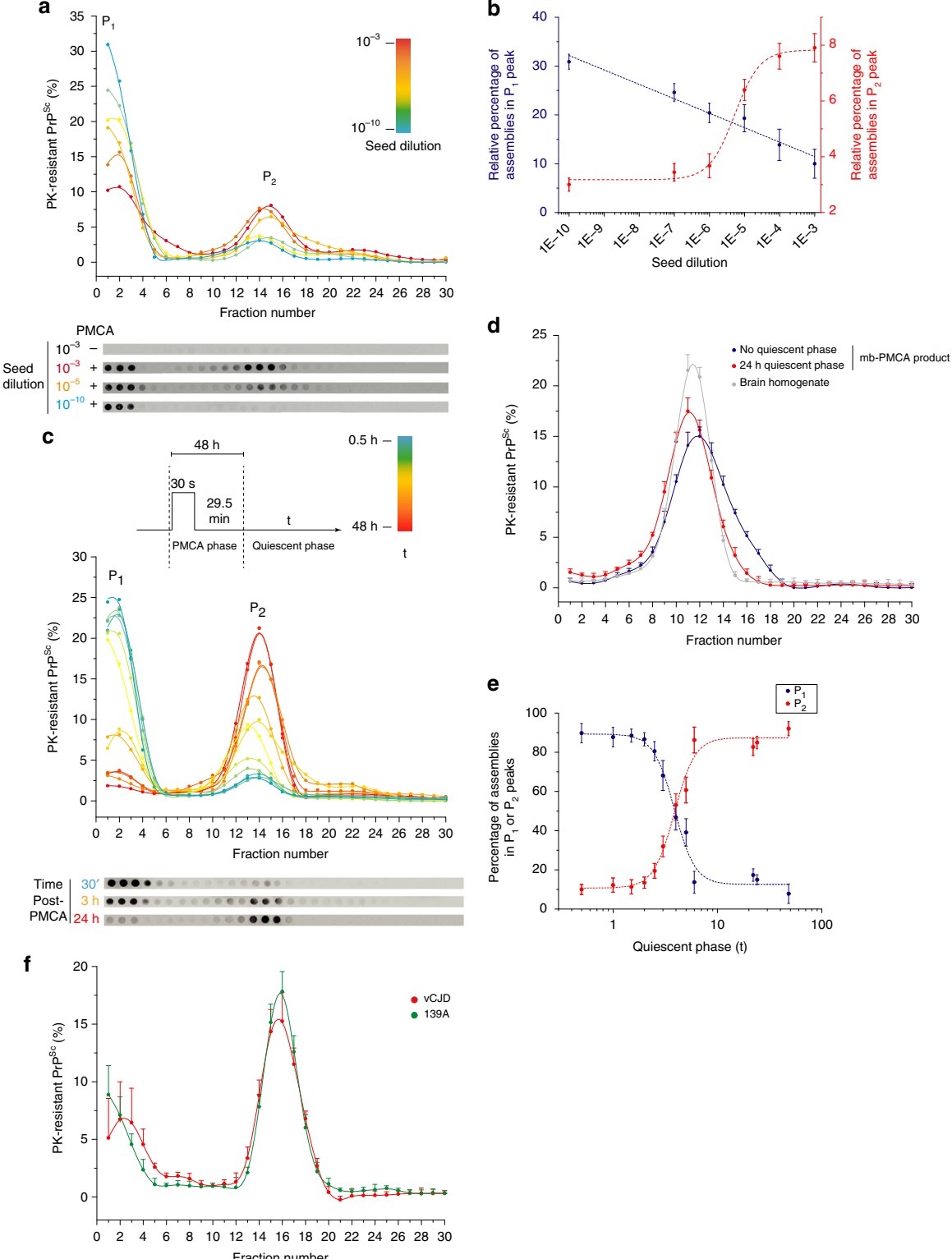

formation of the $P_1$ peak by using high dilutions of the inoculum seed, followed by quiescent incubations (i.e., without sonication) at 37 °C for increasing periods. As shown with 127S prions, the SV analysis at defined incubation time points post-PMCA revealed a time-dependent, drastic decrease in the population of $P_1$ in favor of $P_2$ (Fig. 2c).

Comparing the distribution in isopycnic gradients[20] of the PrP$^{res}$ populations at 0 h and 24 h of quiescent incubation revealed a quasi-similar density for PrP$^{res}$ assemblies composing the $P_1$ and the $P_2$ peaks (Fig. 2d). Thus, the SV increase of $P_2$ compared to $P_1$ results strictly from a quaternary structure rearrangement through size increase.

During the quiescent incubations, the formation of assemblies sedimenting in $P_2$ exhibited a bimodal behavior (i.e., absence of assemblies of intermediate size), without any noticeable shift in the $P_2$ peak position (Fig. 2c). This suggests that the formation of

**Fig. 2** Seed concentration- and time-dependent dynamic evolution of the PMCA-generated PrP$^{Sc}$ assemblies. **a, b** SV profiles of mb-PMCA products seeded with serial ten-fold dilutions from 127S-infected brain homogenates, as indicated. Thirty minutes after the last sonication, the amplified products were solubilized and SV-fractionated. The mean relative levels of PK-resistant PrP$^{Sc}$ per fraction were obtained from the immunoblot analysis of $n = 4$ independent fractionations of PMCA reactions (**a**, representative dot-blot shown). Variation in the $P_1$ and $P_2$ peak maximum (mean ± SEM values) as a function of the logarithm of the seed dilution factor (**b**). **c** PK-resistant PrP$^{Sc}$ sedimentograms from the PMCA products generated with 127S prions ($10^{-5}$ dilution) and further incubated at 37 °C during the indicated quiescent phase (t), i.e., without sonication. At each time point, the collected products were frozen prior SV analysis. All collected samples were then thawed, fractionated in parallel by SV and analysed by immunoblot (**c**, $n = 3$ independent experiments, representative dot-blot shown). **d** PK-resistant PrP$^{Sc}$ isopycnic sedimentograms from PMCA products generated with 127S prions ($10^{-5}$ dilution) and immediately fractionated at the end of the PMCA reaction (blue line and symbol) or after a 24-h-quiescent incubation at 37 °C (red line and symbol). At each time point, the collected samples were frozen. All collected samples were then thawed, fractionated in parallel by sedimentation at the equilibrium[20] and analysed by immunoblot (the mean ± SEM levels of PK-resistant PrP$^{Sc}$ per fraction were obtained from the immunoblot analysis of $n = 3$ independent fractionations of PMCA reactions). As control, the density profile of PK-resistant PrP$^{Sc}$ assemblies from the brain of terminally sick tg338 mice infected with 127S prions (solubilization at 37 °C to mimic the PMCA conditions) is shown (gray line and symbol). **e** Evolution of the percentage of $P_1$ and $P_2$ peak surface areas as a function of the quiescent phase post-PMCA reaction (**c**). **f** PK-resistant PrP$^{Sc}$ sedimentograms from the PMCA products generated with 139 A and vCJD prion seeds ($10^{-5}$ dilution) and further incubated for a quiescent period of 48 h at 37 °C (mean ± SEM values from $n = 3$ independent experiments)

these assemblies resulted from the association with a specific number of assemblies present in $P_1$. Analyzing the time-dependent surface variation of $P_1$ and $P_2$ showed that the formation of $P_2$ assemblies started slowly, increased steadily from ~2–3 h up to ~7 h and finally reached a plateau. This sigmoidal variation is hallmark of an autocatalytic reaction[54] and indicates that the assemblies present in $P_2$ enhance their own formation. Similarly, the 139A and vCJD prions showed a bimodal evolution of $P_1$ to $P_2$ during a 24-h quiescent phase (Fig. 2f), arguing in favor of a generic process of transformation.

To determine whether the $P_2$ peak assemblies could further evolve, we extended the quiescent phase up to 30 days. For the three prion strains, the sedimentogram curves at 7 and 30 days showed a translational shift in the $P_2$ peak to higher fractions, indicative of an isokinetic increase in their mean average sizes (Fig. 3a, left curves). This size translation deeply contrasts with the bimodal phase observed during the 0 to 7-day quiescent incubation and highlights a change in the kinetic regime. This new regime would be compatible with a coalescence process[51–53], whereby assemblies would grow by end-to-end or lateral association rather than by incorporation of monomers (Supplementary Fig. 5).

Altogether, the quaternary structure variation of PrP$^{res}$ assemblies as a function of seed-concentration or time followed two distinct kinetic regimes. The first regime, occurring during the early steps of the conversion process, leads to a bimodal and cooperative size increase, which indicates the existence of an autocatalytic transformation of PrP$^{res}$ assemblies present in $P_1$ to $P_2$. The bimodal aspect of the size distribution tends to indicate that the PrP$^{res}$ assemblies forming $P_1$ structurally differ from those forming $P_2$. The second regime, occurring on long-term quiescence is more compatible with a coalescence process.

**Quasi-irreversible transformation of PrP$^{res}$ from $P_1$ to $P_2$.** The bimodal and cooperative transformation of $P_1$ to $P_2$ reported in Fig. 2c, e is incompatible with the existence of an equilibrium between the assemblies populating these peaks and a coalescence or coagulation process[51–53] (Supplementary Fig. 5). To further disprove the existence of such equilibrium (or detailed-balance), we first set a mb-PMCA regime favoring the formation of the $P_1$ peak together with the $P_2$ peak (low dilution of the inoculum seed, Fig. 4a), isolated by SV the assemblies forming $P_1$ and $P_2$ and studied their quaternary structural evolution on isolation during quiescent incubation for 7 days at 37 °C. Almost all $P_1$ in isolation was transformed into $P_2$ (Fig. 4b). The $P_2$ peak in isolation did not lead to the retro-formation of the $P_1$ peak by depolymerization (Fig. 4c). These observations reflect an

irreversible transformation process and underly the absence of an equilibrium between $P_2$ and $P_1$. The irreversible character of the transformation of $P_1$ to $P_2$ argues in favour of the existence of a thermodynamically driven "locking" process. This implies structural rearrangements of $P_1$ assemblies and formation of higher stable $P_2$ objects.

**$P_1$ and $P_2$ contain structurally distinct PrP$^{res}$ assemblies.** To further confirm the structural rearrangement in the PrP$^{Sc}$ assemblies accompanying the $P_1$ to $P_2$ transformation, we determined the specific infectivity of the $P_1$ and $P_2$ assemblies. A 127S-PMCA product was fractionated at the end of the reaction or after 48 h of quiescent incubation. Pools of fractions corresponding to the $P_1$ and $P_2$ peaks were inoculated into reporter tg338 mice. The specific infectivity (infectivity per PrP molecule), which is mostly associated to PrP$^{res}$ assemblies (i.e., negligible contribution of PK-sensitive PrP$^{Sc}$ species to 127S infectivity[15,20]), was calculated from the mean survival time using 127S dose-response curves[15]. The specific infectivity of the $P_1$ peak assemblies was 50–100-fold higher than that of the $P_2$ peak assemblies (Fig. 5). This indicates that the $P_1$ and $P_2$ peaks contain structurally distinct sets of PrP$^{res}$ assemblies, named $A_i$ and $B_i$ (the i index referring to the number of monomer/subunit in the assembly).

The specific infectivity of $P_2$ did not change over a longer period of quiescent incubation (7 days), suggesting that the transformation of the assemblies present in the $P_2$ peak into larger assemblies was not associated with a structural change measurable by their specific activity.

**Architectural characterization of $A_i$ and $B_i$ assemblies.** To characterize the structural difference between $A_i$ and $B_i$ assemblies at the level of their elementary subunit[41], we used a size exclusion chromatography (SEC) method in native condition, allowing hydrodynamic radius-based analyses. To determine if the hydrodynamic radius from $B_i$ elementary subunit (suPrP$^B$) differ from that of $A_i$ (suPrP$^A$), 127S-PMCA products generated at high-seed dilution ($10^{-8}$) were analysed by SEC immediately at the end of the reaction (defined at $t_0$) or after a 7-day quiescent incubation. At $t_0$, the SEC profile showed the existence of a unique peak eluting at 14.7 ml (Fig. 6a). As PMCA products generated at $10^{-8}$ seed dilution mostly contain $A_i$ assemblies in the $P_1$ peak (Fig. 6b), one can attribute the SEC peak at $t_0$ to suPrP$^A$. After the 7-day quiescence, the chromatogram revealed the emergence of an additional peak eluting at 15.5 ml (Fig. 6a), which correlates with the transformation of $A_i$ to $B_i$ observed by SV (Fig. 6b). This new peak was thus attributed to suPrP$^B$. The

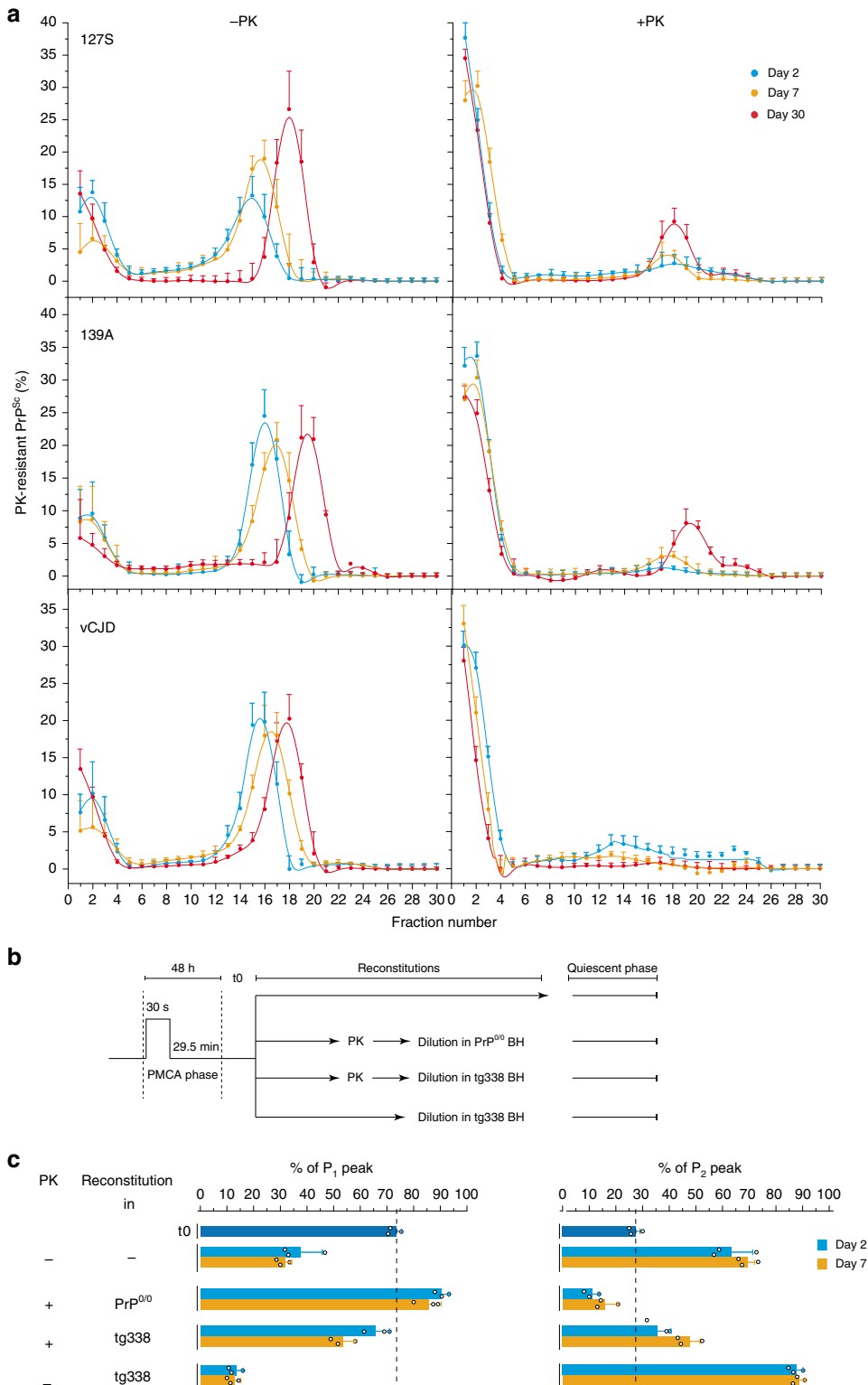

tiny difference observed in the elution volume between suPrP$^A$ and suPrP$^B$ suggests a difference in their hydrodynamic radius (suPrP$^B$ assemblies being more compact than suPrP$^A$ assemblies), and therefore a difference in their structure.

To gain further insight into the mechanism of suPrP$^B$ formation, 127S-PMCA products generated with different seed concentrations as in Fig. 2a were analysed by SEC. At high-seed dilution ($10^{-8}$ dilution factor), the chromatogram revealed the existence of suPrP$^A$ (Fig. 6c). Lower seed dilutions led to the emergence of a new peak

with an elution volume of 15.5 ml corresponding to the emergence of suPrP$^B$, and a shift toward lower elution volume of suPrP$^A$ (Fig. 6c). This last phenomenon could be the chromatographic echo of a dissociation/association equilibrium displacement between different species during the separation on the SEC column[55,56].

Collectively, the SEC analysis of the $P_1$ to $P_2$ transformation demonstrates that the formation of $B_i$ species is concerted with the emergence of a new elementary subunit (suPrP$^B$). suPrP$^B$ differs from suPrP$^A$ by its hydrodynamic radius and therefore its

**Fig. 3** PrP-dependent generation of $B_i$ assemblies from $A_i$ assemblies. **a** PMCA products from 127S, 139A and vCJD prions ($10^5$, $10^4$, and $10^4$ diluted seeds, respectively) were treated with or without PK to eliminate $PrP^C$ before quiescent incubation at 37 °C for 2 days, 7 days or 30 days, as indicated. At each time point, the collected products were frozen. All collected samples were then thawed, SV-fractionated in parallel and analysed by immunoblotting (mean ± SEM values from $n = 3$ independent experiments). **b**, **c** Relative percentage of $P_1$ versus $P_2$ peaks in SV-sedimentograms from ±PK-treated PMCA products reconstituted in $PrP^{0/0}$ or $PrP^C$ containing tg338 mouse brain homogenates, and incubated in quiescent conditions for 2 or 7 days. **b** PMCA products were generated with a $10^5$-diluted 127S prion seed. At the end of the PMCA reaction (t0), the products were mixed, and eventually treated with high concentration of PK to remove residual $PrP^C$. After PK inhibition, the products were then diluted 1:1 in either $PrP^{0/0}$ brain homogenate or in tg338 brain homogenate and incubated for 2 days or 7 days at 37 °C in quiescent conditions. **c** The PMCA products were then fractionated by sedimentation velocity and analysed for $PrP^{Sc}$ content by immunoblot. The amount of $PrP^{Sc}$ in the fractions corresponding to $P_1$ and $P_2$ populations was quantified. The results shown are the mean ± SEM values from $n = 3$ independent experiments

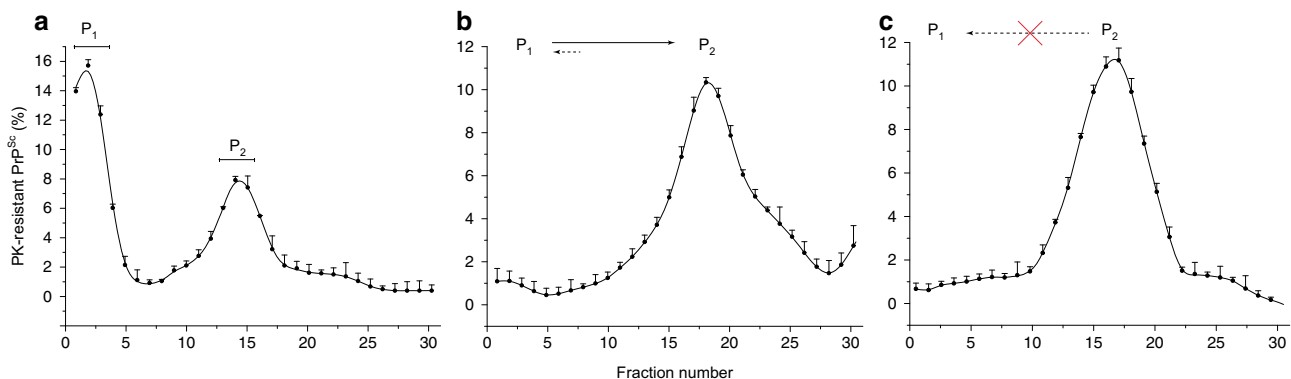

**Fig. 4** Quaternary structural evolution of isolated $PrP^{Sc}$ assemblies in $P_1$ and $P_2$ peaks on quiescent incubation. **a** SV profile of PMCA products seeded with $10^6$-diluted 127 S brain homogenate, leading to the formation of $P_1$ and $P_2$ assemblies (as in Fig. 2a). The fractions corresponding to $P_1$ and $P_2$ peaks were pooled as indicated, and further incubated for 7 days at 37 °C in quiescent conditions, prior SV analysis. **b** On quiescent incubation, most of the assemblies present in the pooled $P_1$ fractions evolved and formed $P_2$. **c** On quiescent incubation, the pool of $P_2$ fractions did not evolve, underlying the irreversible character of the $P_1$ to $P_2$ transformation and the absence of an equilibrium between $P_1$ and $P_2$. The results shown are the mean ± SEM values from $n = 3$ independent fractionations

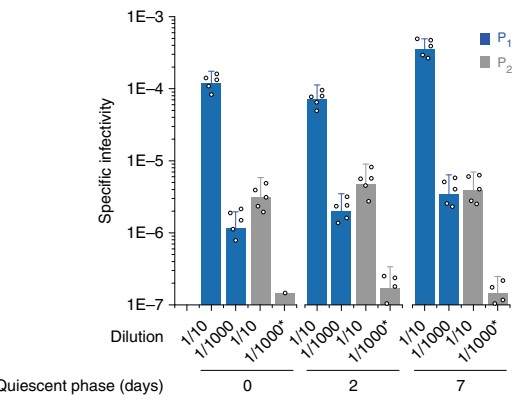

**Fig. 5** Specific infectivity of the $P_1$ and $P_2$ peaks post-PMCA reaction and after quiescent incubation. Fractions corresponding to $P_1$ (fractions 1–3) and $P_2$ (fractions 14–16 (days 0 and 2) or 16–18 (day 7)) from PMCA products seeded with $10^6$-diluted 127 S brain homogenate were pooled and inoculated into groups of reporter tg338 mice at two different dilutions (1:10 and 1:1000) for better accuracy. The specific infectivity of the assemblies was calculated from the mean survival time of the mice using a 127S dose-response curve. Asterisk: incomplete attack rate

structure. The structural difference between $suPrP^A$ and $suPrP^B$ is at the origin of their physicochemical properties and their aggregation propensity. The existence of conformationally distinct suPrP further demonstrates that $A_i$ and $B_i$ are fundamentally different in terms of ultrastructure.

**The formation of $B_i$ from $A_i$ assemblies requires $PrP^C$.** Our previous studies revealed that only ∼30% of the $PrP^C$ substrate is converted into $PrP^{Sc}$ after a complete round of mb-PMCA[45,46]. To determine whether the remaining 70% still participated in the transformation of $A_i$ to $B_i$ assemblies during the quiescent phase, PMCA products from the 139A, 127S and vCJD prions were treated with PK to eliminate $PrP^C$ before quiescent incubation at 37 °C. As shown in Fig. 3a (right panel), the amount of $B_i$ assemblies generated during the 48-h quiescent incubation was drastically and invariably decreased. Further quiescent incubation for 7 and 30 days in the absence of $PrP^C$ allowed the formation of comparatively low amounts of $B_i$ assemblies for 127S and 139A prions.

To determine if the drastic decrease of $A_i$ to $B_i$ transformation was specific to depletion of $PrP^C$ or of cofactors, we performed reconstitution experiments of ±PK-treated 127S-PMCA products (e.g., without potential, PK-susceptible co-factors) with either $PrP^{0/0}$ brain homogenate (e.g., media containing all brain cofactors except $PrP^C$) or tg338 brain homogenate (e.g., media containing all cofactors and $PrP^C$) before 48 h or 7 days of quiescent incubation (Fig. 3b). The quiescent products were then SV-fractionated and the amount of $PrP^{Sc}$ in the fractions corresponding to $P_1$ and $P_2$ peaks was quantified. As shown in Fig. 3c, reconstitution of the PK-treated PMCA amplicons with $PrP^{0/0}$ brain homogenate did not allow $B_i$ neoformation as compared with reconstitution in tg338 media. A depolymerization of $B_i$ assemblies was even observed when the reconstitution was done in $PrP^{0/0}$ brain homogenate. Thus, the contribution of PK-sensitive PrP conformers and protein cofactors appeared

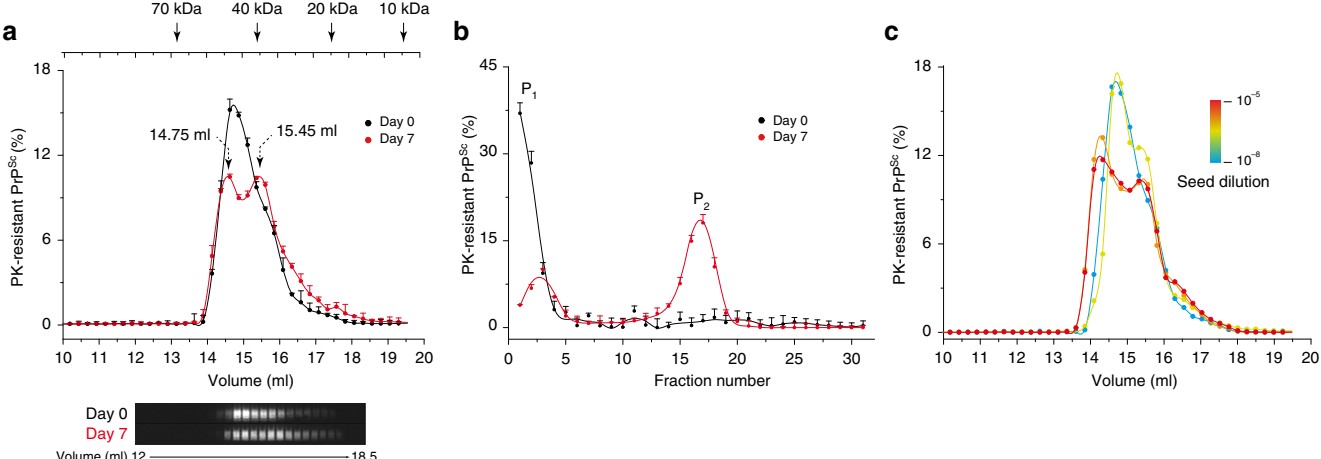

**Fig. 6** Characterization of the elementary subunit of PMCA-generated PrP$^{Sc}$ assemblies by size exclusion chromatography (SEC) under native conditions. **a** SEC analysis ($n \geq 3$ independent fractionations) of mb-PMCA products generated with 127 S prions ($10^{-8}$ dilution) immediately after the PMCA phase (day 0) or after 7 days of quiescent incubation (day 7). A representative immunoblot corresponding to elution volumes 12 ml to 18 ml is shown (electrophoretic concentration on loading as in Supplementary Fig. 1). The column calibration was performed using standard MW calibrants under identical conditions as for PMCA products analysis. **b** Representative sedimentogram of mb-PMCA products generated with 127 S prions ($10^{-8}$ dilution) post-PMCA reaction (day 0) and after a 7-day quiescent incubation, highlighting the $P_1$ to $P_2$ evolution of PrP$^{Sc}$ assemblies. **c** SEC profiles of mb-PMCA products generated with 127S seeds at different dilution factors, as indicated. Thirty minutes after the last sonication, the amplified products were solubilized and SEC-fractionated. The mean relative levels of PK-resistant PrP$^{Sc}$ per fraction were obtained from the immunoblot analysis of $n = 3$ independent fractionations of PMCA reactions. Note the formation of at least two distinct set of assemblies as function of seed concentration

negligible. The formation of $B_i$ assemblies upon reconstitution of PK-treated mb-PMCA product with tg338 brain homogenate also indicated that the N-terminal segment of $A_i$ had a low contribution to the process.

Finally, the importance of PrP$^C$ in the $A_i$ to $B_i$ transformation was further strengthened when comparing the quiescent evolution of non-PK-treated mb-PMCA products freshly reconstituted with tg338 brain homogenate with that of mb-PMCA products alone. As can be seen, the amount $B_i$ assemblies formed was ~1.4-fold increased upon fresh reconstitution (Fig. 3c).

Collectively, this set of reconstitution experiments indicates that the $A_i$ to $B_i$ transformation can be qualified as a pure PrP$^C$-dependent process. PrP$^C$ requirement suggests that $B_i$ assemblies result from the integration/conversion of PrP$^C$ into $A_i$ assemblies during the quiescent phase. The appearance of a low amount of $B_i$ after a long incubation period without PrP$^C$ may result from the leakage of monomers from a conformer cosedimenting with $A_i$.

**Kinetic scheme describing the transformation of $A_i$ to $B_i$.** To establish a kinetic mechanism and provide a molecular interpretation of the assemblies dynamics during the quiescent phase, a number of elementary steps was identified based on experimental observations and used as unavoidable constraints[57]. The first constraint is the existence of two structurally distinct PrP$^{Sc}$ subassemblies, namely $A_i$ and $B_i$, with distinct dynamics. Indeed, structurally equivalent assemblies would fail to present a bimodal size distribution, cooperative seed concentration and kinetic evolution or distinct specific infectivity, as mathematically demonstrated in Supplementary Fig. 5. The second constraint is the existence of a detailed-balance between the PrP$^{Sc}$ assemblies and their elementary subunit, as previously shown[41], making the size distribution of the PrP$^{Sc}$ assemblies highly dynamic and dependent on the assembly concentration (Fig. 1e). The 3rd constraint is that $A_i$ and $B_i$ assemblies are in detailed-balance with their respective suPrPs (Fig. 6, denoted $suPrP^A$ and $suPrP^B$) but with distinct equilibrium constants $K_{eq}^A$ and $K_{eq}^B$. Thus, at any moment of the process of assembly transformation of $A_i$ to $B_i$, the

following equilibrium should be respected:

$$A_i \rightleftharpoons A_{i-1} + suPrP^A \quad (1)$$

$$B_i \rightleftharpoons B_{i-1} + suPrP^B \quad (2)$$

The equilibrium constants $K_{eq}^{Ai}$ and $K_{eq}^{Bi}$ govern the respective size distribution of the $A_i$ and $B_i$ assemblies and, thus, the bimodal aspect of the curve. According to our previous SV calibrations with PrP oligomers and globular mass markers[15], the size distribution of the $A_i$ and $B_i$ subassemblies were fixed: $i_A < 5$ and $i_B$ centered around 20 PrP-mers. Due to the limited resolution of SV fractionation for small assemblies, we assumed that $A_i$ and $suPrP^B$ cosedimented. The fourth constraint relies on the fact that A to B transformation requires PrP$^C$ and that the kinetic is cooperative (Figs. 1e and 2). This cooperativity implies that B subassemblies facilitate their own formation according to an autocatalytic process. This can be resumed by the following minimalistic autocatalytic process:

$$suPrP^A + suPrP^B \rightleftharpoons C \quad (3)$$

$$C + PrP^C \rightarrow 2suPrP^B \quad (4)$$

where C is an active complex reacting with PrP$^C$ that generates B assemblies. Considering that suPrP$^B$ can condense into $B_2$[41] and according to detailed-balance (2), one can establish the reaction model describing the formation of $B_i$ assemblies from the neo-formed $suPrP^B$:

$$2suPrP^B \rightleftharpoons B_2 \quad (5)$$

Altogether, these five elementary steps constitute the reaction mechanism that describes the transformation of $A_i$ into $B_i$ subassembly species.

To validate the designed mechanism, we translated these elementary reactions into time-dependent differential equations (for more details, see Supplementary Note) and performed kinetic simulations using the size distribution of the PrP$^{Sc}$ assemblies immediately after cyclic amplification as the initial condition (blue curve in Fig. 2a). According to the model, the simulated size

distribution variation as a function of time showed bimodal behavior, as was experimentally observed (Fig. 7a). Furthermore, the theoretical size distribution centroid presented similar sigmoidal patterns to those of the experimental data (Fig. 7b), arguing in favor of an autocatalytic kinetic model describing the overall quaternary structure evolution of PrP$^{Sc}$ assemblies during the quiescent phase. The analysis of the model (for more details, see Supplementary Note) revealed that the autocatalytic formation of $B_i$ species occurs at the expense of $A_i$ species and with PrP$^C$ consumption (Fig. 7c). According to this model, when PrP$^C$ is in large excess, $A_i$ constitutes the limiting compound for the formation of $B_i$ assemblies. Therefore, during the quiescent phase, the PrP$^C$ to PrP$^{Sc}$ conversion rate is directly proportional to the amount of $A_i$ assemblies (Fig. 7c).

## Discussion

The mechanisms of prion replication and the dynamics responsible for prion structural diversification in the infected host remain unclear and rarely addressed. In the actual framework of the prion paradigm, the templating process occurs at the prion assembly interface, leading to an increased size of the complex formed by the template:substrate, out of the fragmentation/depolymerization context. The atypical size distribution observed here at the early replication stage for three distinct prion strains, where accumulation of small-sized assemblies dominates, contrasts with this canonical templating model and requires an additional process that considers the replication dynamics.

As shown in vivo for the vCJD, 127S and 139A strains, the early stage of the replication process in the brain is dominated by the accumulation of small assemblies, whereas higher-size subsets are detected at the terminal stage of pathogenesis. Such quaternary structural diversity, and beyond the existence of structurally distinct types of assemblies, as defined by their specific infectivity (refs. [15,20] and Supplementary Fig. 6), can be exclusively explained by the existence of a balance between at least two kinetic modes taking place at different stage of the pathogenesis. Both can be governed by evolution or a fluctuation in the replication microenvironment due to the physio-pathological state of the infected animal and/or to the sequential involvement of specific prion-replicating cell types. However, another possibility lies in the intrinsic and deterministic properties of the PrP replication process to generate structurally distinct types of assemblies. Discriminating between these two non-mutually exclusive hypotheses is technically difficult in vivo. The mb-PMCA as a bona fide amplification method in a more simplified and kinetically controlled context constitutes a relevant method for investigating the intrinsic propensity of the replication process to generate structurally distinct assemblies. Interestingly, and against common belief, the size distribution of the PrP$^{Sc}$ assemblies used as seeds was relatively insensitive to repeated sonication cycles when a simple dilution displaced the assemblies towards a smaller size (Fig. 1e and ref. [41]). These observations exclude the contribution of the fragmentation process during the mb-PMCA sonication cycles to the size distribution pattern of PrP$^{Sc}$ assemblies and emphasize the existence of a constitutional dynamic between the PrP$^{Sc}$ subpopulation[41], which should be considered during the replication process. We showed that two sets of PrP$^{Sc}$ assemblies, $A_i$ and $B_i$, were generated during the mb-PMCA reaction. The $A_i$ and $B_i$ assemblies constitute two structurally distinct PrP$^{Sc}$ subpopulations. Beside the fact that the bimodal size distribution instead of a continuum constitutes an indirect but solid argument for structural differences in the PrP assemblies populating the $P_1$ and $P_2$ peaks, the best arguments are undoubtedly their distinct specific infectivity and the existence of two distinct elementary subunits. The irreversibility of the $P_1$ to

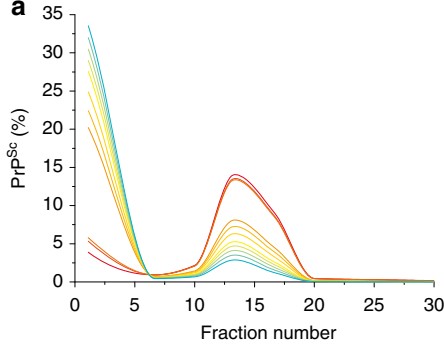

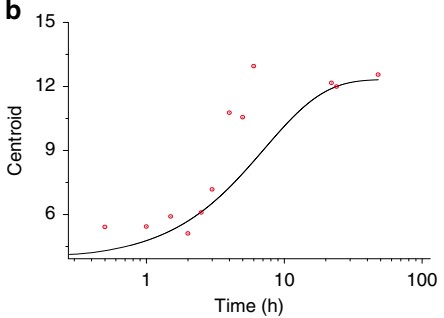

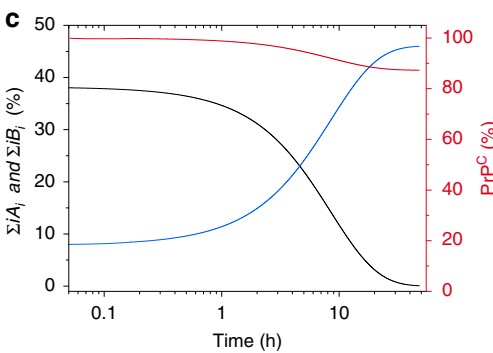

**Fig. 7** Mathematical modeling of the time-dependent dynamic evolution of the PMCA-generated PrP$^{Sc}$ assemblies. **a** The size distribution evolution of a structurally distinct set of assemblies $A_i$ and $B_i$ dimensioned on gradient fraction numbers was simulated based on the kinetic scheme described in the results section (Eq. 1 to 5) and the Supplementary note. **b** The time dependency evolution of the simulated centroid (black line) and centroid calculated from experimental sedimentograms of Fig. 2d (red circle) show a similar shape, supporting the cooperativity hypothesis of the transformation of $A_i$ into $B_i$. **c** The simulation of time dependency evolution of the total amount of $A_i$ assemblies ($\sum iA_i$ in black), $B_i$ assemblies ($\sum iB_i$ in blue) and the monomer (in red) revealed that $A_i$ assemblies constitute the limiting species for the conversion of PrP$^C$ during the quiescent phase. In the present simulation framework (for more details, see Supplementary note), only 14% of PrP$^C$ is consumed

$P_2$ transformation (Fig. 4) physically demonstrates a PrP structural rearrangement associated to the $A_i \rightarrow B_i$ transformation process. Therefore, the prion replication process per se intrinsically generates structurally diverse PrP$^{Sc}$ subassemblies in a deterministic process.

According to our SV experiments, small-sized PrP$^{Sc}$ assemblies were mainly formed at the early stage of prion replication in the brain and during the mb-PMCA reaction. This was observed with three distinct prion strains on three different PrP genetic backgrounds. Considering that the PrP$^{Sc}$ assemblies that constitute each strain are structurally distinct, one can ask how distinct PrP$^{Sc}$ assemblies can all generate $A_i$ assemblies that harbor strain

structural information while showing the same quaternary structure (at the SV resolution). The first explanation can be the existence of a common narrow subpopulation of PrP$^{Sc}$ (with respect to their quaternary structure) within the three strains that serves as the best replicator and participates in the formation of A$_i$ assemblies. However, the PrP$^{Sc}$ quaternary structure subset that exhibits the highest specific infectivity in vivo (i.e., the best replicator) can be associated with either small-size assemblies (i.e., 127S and 139A[15,20] and Supplementary Fig. 6a, respectively) or high-molecular-weight assemblies (i.e., vCJD, Supplementary Fig. 6b) and is therefore strain-dependent. The existence of a structurally common PrP$^{Sc}$ subpopulation is thus unlikely to be at the origin of the generic formation of a small-size subset in the brain or A$_i$ assemblies in the mb-PMCA condition. Intrinsically, the early steps of the replication process favor the emergence of mainly one subspecies A$_i$ with a highly narrowed size distribution, arguing in favor of a quaternary structural convergence phenomenon during this phase. This structural convergence concerns the PrP domain that governs polymerization (the size of assemblies). As the A assemblies harbor the strain structural determinant, the A$_i$ assemblies would present a certain degree of structural variability, allowing strain structural information encoding.

All along the quiescent phase and for the three prion strains studied, the A$_i$ assemblies constitute the precursor species in the formation of B$_i$ assemblies. Furthermore, there is compelling evidence that the presence of PrP$^C$ is required for the evolution of A$_i$ into B$_i$ assemblies. The set of reconstituted media experiments (Fig. 3) led us to firmly exclude the contribution of PK-susceptible cofactors and highlighted the existence of a secondary templating pathway. In addition, the N-terminal part of PrP$^{Sc}$ (at least for 127S seeds) is dispensable for $A_i \rightarrow B_i$ transformation. Even if the yield of the process was decreased when 127S PMCA products were PK-treated (i.e., removal of the N-terminal domain) before reconstitution and quiescent incubation with tg338 normal brain homogenate, this could clearly be attributed to the kinetic effect of the two-fold dilution factor of both PrP$^C$ and PMCA product occurring during the reconstitution.

According to the kinetic model describing the autocatalytic formation of B$_i$ during the quiescent phase, A$_i$ is the limiting species for conversion when large amounts of PrP$^C$ are present (Fig. 7c and Supplementary note). The cooperative disappearance of P$_1$ in favor of P$_2$ strongly suggests an autocatalytic process for the transformation of A$_i$ to B$_i$ (reactions 3 and 4). This last phenomenon shows the existence of a secondary autocatalytic process, undescribed to our knowledge until now, in the canonical prion replication process[32]. It can be reasonably envisaged that A$_i$ have the intrinsic propensity to generate B$_i$ assemblies in the presence of PrP$^C$ assemblies with a very low efficiency. This parallel pathway to the autocatalytic process can then explain how the first set of B$_i$ assemblies is generated (Fig. 8).

The existence of a secondary autocatalytic process can be a way to maintain PrP$^{Sc}$ structural diversity throughout the evolution of the pathology. In the absence of this secondary autocatalytic process, the system only selects the best replicator assembly. Here, the best replicator is A$_i$ assembly according to its specific infectivity (Fig. 5). The secondary templating pathway allows the system to escape this rule, leading the accumulation of the autocatalytic pathway product (here, the B$_i$ assemblies). This phenomenon can explain why, for certain prion strains, the most infectious assemblies represent a minor population, while those with the lowest specific infectivity mostly accumulate[15,20].

The deterministic aspect of the replication process to generate a structurally diverse set of assemblies contrasts with the widespread idea that considers the prion diversification process within a given strain (often referred to as the creation of prion quasi-species) as a stochastic event and as a process that is governed by environmental fluctuations[9]. The secondary autocatalytic pathway leading to the formation of B$_i$ subassemblies can participate in prion adaptation during transmission events with species barriers. Considering that the transmitted inoculum initially contains A$_i$ and B$_i$ assemblies, the autocatalytic conversion process of B$_i$ can kinetically drive the adjustment and integration of the new-host PrP$^C$ to generate host-adapted B$_i$ assemblies.

## Methods

**Ethics**. Animal experiments were conducted in strict compliance with ECC and EU directives 86/009 and 2010/63 and were approved by the local ethics committee of the author's institution (Comethea; permit numbers- 12/034 and 15/045).

**Transgenic mouse lines and prion strains**. The ovine (tg338 line; Val136-Arg154-Gln171 VRQ allele), human (tg650 line; Met129 allele) and mouse (tga20) PrP transgenic lines have been described previously[29,42,43]. The mouse lines were homozygous and overexpressed approximately 8-, 6-, and 10-fold amounts of heterologous PrP$^C$ on a mouse PrP-null background. PrP$^{0/0}$ mice were the so-called Zürich-I mice[58]. Cloned 127S scrapie, human vCJD and mouse 139A prion strains were serially passaged in tg338, tg650, and tga20 mice, respectively[45,46]. These strains were used as pools of mouse-infected brains and prepared as 20% wt/vol homogenates in 5% glucose by use of a tissue homogenizer (Precellys 24 Ribolyzer; Ozyme, France).

**Time course analysis of prion accumulation**. Eight-week-old female tg338, tg650 and tga20 mice were inoculated intracerebrally in the right cerebral hemisphere with 127S, vCJD or 139A prions (20 µl of a 10% brain homogenate dose). Infected animals were euthanized by cervical column disruption in triplicate at regular time points and at the terminal stage of disease. Brains were removed and kept for PrP$^{Sc}$ size fractionation.

**Miniaturized bead-PMCA assay**. The miniaturized bead-PMCA (mb-PMCA) assay[16,41,46] was used to amplify prions. Briefly, serial ten-fold dilutions of 127S, vCJD, and 139A prions (mouse brain homogenates diluted in PMCA buffer) were mixed with brain lysates (10% wt/vol) from healthy tg338, tg650 and tga20 mice as respective substrates and subjected to one round of 96 cycles of 30-s sonications (220–240 Watts) followed by 29.5 min of incubation at 37 °C. With a >10$^4$ dilution of the seeds, input PrP$^{Sc}$ is not detected in the mb-PMCA products. PMCA was performed in a 96-well microplate format (Axygen, Corning) using a Q700 sonicator (QSonica, USA, Delta Labo, Colombelles, France). For quiescent incubation, the samples were left in the incubator at 37 °C for the indicated period of time, without any sonication. To eliminate residual PrP$^C$ present in the PMCA products before quiescent incubation, the samples were treated with PK (80 µg/ml final concentration). The treatment was stopped by adding 2 mM Pefabloc and 1x EDTA-free protease inhibitor cocktail. All final products were kept for PrP$^{Sc}$ size fractionation, and aliquots were PK-digested (115 µg/ml final concentration, 0.6% SDS, 1 h, 37 °C) prior to immunoblot analyses, as described below.

For reconstitution experiments, mb-PMCA products were generated with a 10$^5$-diluted 127S prion seed. At the end of the mb-PMCA reaction, the products were mixed, eventually treated with PK (150 µg/ml final concentration, 1 h, 37 °C). PK activity was inactivated by the combined addition of 4 mM Pefabloc and 2x EDTA-free protease inhibitor cocktail. The products were then diluted 1:1 in either PrP$^{0/0}$ or in tg338 brain homogenate and incubated for 48 h or 7 days at 37 °C in quiescent conditions. The PMCA products were then fractionated by sedimentation velocity and analysed for PrP$^{Sc}$ content by immunoblot.

**Sedimentation velocity fractionation**. SV experiments were performed as described previously[15,20,41]. Mouse brain homogenates or PMCA products were solubilized by adding an equal volume of solubilization buffer (50 mM HEPES pH 7.4, 300 mM NaCl, 10 mM EDTA, 4% wt/vol dodecyl- β-D-maltoside (Sigma)) and incubated for 45 min on ice. Sarkosyl (N-lauryl sarcosine; Fluka) was added to a final concentration of 2% wt/vol, and the incubation continued for an additional 30 min on ice. A total of 150 µl of solubilized samples was loaded atop a 4.8-ml continuous 10–25% iodixanol gradient (Optiprep, Axys-Shield), with a final concentration of 25 mM HEPES pH 7.4, 150 mM NaCl, 2 mM EDTA, 0.5% Sarkosyl. The gradients were centrifuged at 285,000 g for 45 min in a swinging-bucket SW-55 rotor using an Optima LE-80K ultracentrifuge (Beckman Coulter). Gradients were then manually segregated into 30 equal fractions of 165 µl from the bottom using a peristaltic pump and analysed by immunoblotting or bioassay for PrP$^{Sc}$ or infectivity, respectively. To avoid any cross-contamination, each piece of equipment was thoroughly decontaminated with 5 N NaOH followed by several rinses in deionized water after each gradient collection[20].

**Isopycnic sedimentation**. The entire procedure was performed as described previously[20]. Mouse brain homogenates or PMCA products were solubilized as

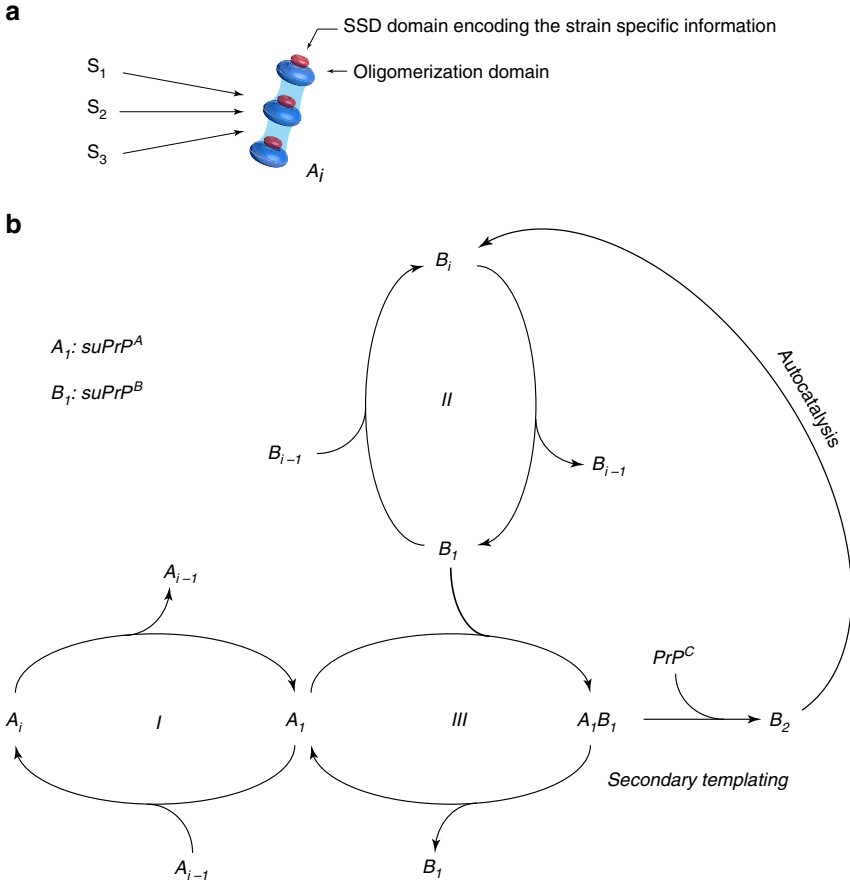

**Fig. 8** Quaternary structural convergence and secondary autocatalytic pathway at the root of the formation of $B_i$ assemblies. **a** Different prion strains ($S_1$, $S_2$, and $S_3$) give rise to the formation of common oligomeric assemblies, termed $A_i$, with a narrowed size distribution during mb-PMCA reactions. This common quaternary structural convergence at the early stage of the replication process suggests the existence of a common conversion pathway and a common oligomerization domain that is independent of the strain structural determinant (SSD, i.e., the PrP domain(s) harboring the replicative and strain information[41,50], represented in red). **b** $A_i$ and $B_i$ assemblies are in an equilibrium/detailed-balance with their respective suPrP (step I and II) as was previously showed[41] and also demonstrated by the dilution experiment (see Fig. 1e). Based on the constraints imposed by the experimental observations, the best model to account for the cooperative and PrP$^C$ dependency transformation of $A_i$ into $B_i$ assemblies implicates the formation of complex between suPrP$^A$ and suPrP$^B$ (step III). The formation of this complex is at the origin of a secondary templating pathway where the transformation of suPrP$^A$ ($A_1$) to suPrP$^B$ ($B_1$) is assisted by suPrP$^B$, making the process autocatalytic

described above. For mouse brain homogenates, solubilization was performed at 37 °C to mimic PMCA conditions. A total of 220 µl of solubilized material was mixed to reach 40% iodixanol, 25 mM HEPES pH 7.4, 150 mM NaCl, 2 mM EDTA, 0.5% Sarkosyl final concentration and loaded within a 4.8 ml of 10–60% discontinuous iodixanol gradient with a final concentration of 25 mM HEPES pH 7.4, 150 mM NaCl, 2 mM EDTA, 0.5% Sarkosyl. The gradients were centrifuged at 115 000 g for 17 hours in a swinging-bucket SW-55 rotor using an Optima LE-80K ultracentrifuge (Beckman Coulter). Gradients were then manually segregated into 30 equal fractions of 165 µl from the bottom using a peristaltic pump and analysed for PrP$^{Sc}$ content by immunoblotting.

**Size exclusion chromatography**. SEC analysis was performed using an ÄKTA-100 purifier FPLC (GE Healthcare). 200 µl of the PMCA products were mixed with an equal volume of 2X-buffer to reach 25 mM HEPES pH 7.4, 150 mM NaCl, 10 mM EDTA, 5 mM n-Dodecyl β-D-Maltoside, 2 % w/w Sarkosyl and 0.5% Triton-X100 final concentration. After centrifugation at 10 000 g for 3 min (no visible pellet), the solution was loaded on Superdex 200 10/300 GL column (24 ml, GE healthcare). The chromatography running buffer was HEPES 25 mM pH7.2, 200 mM NaCl, without detergents to avoid the formation of micellar structure. The flow rate was fixed at 0.35 ml/min. After sample injection, the flow-through of the column was fractionated every 250 µl. The PrP levels per fraction were estimated by western blotting, as for SV. For molecular weight estimation, the Superdex 200 was calibrated with blue dextran molecules with varying molecular weight.

**Analysis of PrP$^{Sc}$ content by immunoblotting**. Aliquots of the SV-fractionated samples were treated with PK (50 µg/ml final concentration, 1 h, 37 °C) before

mixing in Laemmli buffer and denaturation at 100 °C for 5 min. The samples were run on 12% Bis-Tris Criterion gels (Bio-Rad, Marne la Vallée, France) and electrotransferred onto nitrocellulose membranes. In some instances, denatured samples were spotted onto nitrocellulose membranes using a dot-blot apparatus (Schleicher & Schuell BioScience (Whatman)). Nitrocellulose membranes were probed for PrP with 0.1 µg/ml biotinylated anti-PrP monoclonal antibody Sha31[59]. Immunoreactivity was visualized by chemiluminescence (ECL, Pierce or Clarity, Bio-Rad). The protein levels were quantified with ImageLab software after acquisition of chemiluminescent signals with a Chemidoc digital imager (Bio-Rad, Marnes-la-Coquette, France). For all SDS-PAGE analyses, a fixed quantity of human recombinant PrP was employed for consistent calibration of the PrP signals in different gels.

To improve the sensitivity of the western blot detection method for the samples containing low levels of PrP$^{pres}$ (e.g., samples at early stage of replication and SEC fractions) a double-deposit was made to electro-concentrate the sample. Typically, after a first round of sample loading in SDS-PAGE wells, a short migration time was performed to allow running within the acrylamide gel for 2 mm. Then, a second round of sample loading was done identically to the first one and the migration was continued until the front reached 3 cm within the gel. The electrotransfer and detection was then identical as above.

**Bioassays**. The pool of fractions of interest was extemporarily diluted ten-fold in 5% glucose and immediately inoculated via the intracerebral route into reporter tg338 mice (20 µl per pool of fraction, $n = 5$ mice per pool). Mice showing prion-specific neurological signs were euthanized at the end stage. To confirm prion disease, brains were removed and analysed for PrP$^{Sc}$ content using the Bio-Rad

TsSeE detection kit[17] prior to immunoblotting, as described above. The survival time was defined as the number of days from inoculation to euthanasia. To estimate what the difference in mean survival times means in terms of infectivity, strain-specific curves correlating the relative infectious dose to survival times were used, as previously described[15].

**Kinetic simulation**. The details of the kinetic simulation are reported in the Supplementary note. Briefly, two distinct sets of assemblies were considered ($A_i$ and $B_i$). Based on experimental observations, a set of constraints was retained to build biochemical reactions describing the evolution of the quaternary structure of PrP^pres assemblies. The ordinary differential equations of the biochemical reactions 1–6 (in the manuscript) were established and coded in MATLAB for simulations.

**Statistics and reproducibility**. The SV and SEC experiments reported here result from at least three independent experiments (biological repeats). For SV fractionation of in vivo brain material, pools of three brains collected at different time points were used to minimize individual brain-to-brain variations. At least three independent fractionations were done using these pools. For fractionation of in vitro material, a new batch of mb-PMCA amplified products was prepared for each fractionation. The 127 S mouse bioassay was done with $n = 5$ mice per group at two different dilutions to ensure statistical significance with this *fast* prion model, while keeping the number of mice as low as possible for ethical reason. The results are reported as mean values ± standard error of the mean (SEM).

**Reporting summary**. Further information on research design is available in the Nature Research Reporting Summary linked to this article.

## Data availability
The SV and SEC raw data set have been deposited to the DRYAD depository https://doi.org/10.5061/dryad.f28b4m6[60]. All other data are available from the corresponding authors on request.

## Code availability
The MATLAB code associated with this study is available from authors upon request.

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

## Acknowledgements
We thank the staff of the Animal facility (INRA-UEAR, Jouy-en-Josas) for animal care. This work was supported by grants from the Fondation pour la Recherche Médicale (Equipe FRM DEQ20150331689), the European Research Council (ERC Starting Grant SKIPPERAD, number 306321), and the Ile de France region (DIM MALINF).

## Author contributions
A.I.E., F.L., Mo.M., J.B., M.D., H.R. and V.B. conceived and designed the experiments. A.I.E., F.L., Mo.M., J.B., Ma.M., T.K., L.H., F.R., C.J.D., M.D., H.R. and V.B. performed the experiments and analysed the data. A.I.E., H.R. and V.B. wrote the paper. All authors reviewed and edited the paper.

## Competing interests
The authors declare no competing interests.
