## [Peer Review File · Communications Biology]

Reviewers' comments:

Reviewer #1 (Remarks to the Author):

The authors provide a combination of empirical evidence and mechanistic modeling to provide novel insights into PrPSc assembly. The work is extensive, multifaceted, competently performed and quite thought-provoking. It should be of significant interest to the prion field and beyond.

Questions/Comments/suggestions:

1. There are a number of odd word choices that might be reconsidered:

L116: invalidated

L165, L189 and elsewhere: surface; do the authors mean area under the curve?

L201 and elsewhere: kinetical \diamond kinetic

L395: favorized \diamond favored

2. L129 & Fig 1: Need to show/describe SV analysis of PMCA seeded with normal brain.

3. L161: Similarly, what happens to P1 with a normal brain seed in mb-PMCA in this experiment?

4. L256: If Bi is bigger, why does it elute more slowly from SEC?

5. L 261, Fig 6: Why might the high-seed reaction products bifurcate into peaks that are elute both faster and slower than the low-seed products? These results, on the face of it, would suggest that smaller and larger particles are generated with high seed. How then might the smaller particles relate to suPrPA? Might this be another elementary subunit?

6. L296: Fig 3C: This fold increase seems large. My understanding is that this statement is based on the comparison of % P2 peak the PK- tg338 row with that of the PK- no-reconstitution row. The fold increase in the former is more like 1.2 fold. Perhaps I am misunderstanding what is meant here.

7. L300-302: The authors state: "The appearance of a low amount of Bi after a long incubation period without PrPC may result from the leakage of monomers from a conformer cosedimenting with Ai." Might this have something to do with the smaller peaks in Fig 6A and C?

Reviewer #2 (Remarks to the Author):

The manuscript entitled "Quaternary structural convergence and structural diversification of prion assemblies at the early replication stage" (COMMSBIO-19-0516-T) by Igel-Egalon et al. describes quaternary structural conversions in in vivo and in vitro models of prion replication.

The misfolding of the prion protein from the cellular PrPc to the disease-associated PrPsc is thought to progress through oligomeric intermediate states. The dynamic and temporary nature of protein oligomers makes them a difficult research subject in general. In this manuscript the authors describe a technically and conceptionally original approach to investigate the aggregation state and conformation of prion protein oligomers during in vivo and in vitro conversions. By using sedimentation velocity ultracentrifugation approaches the authors discovered two distinct oligomeric populations termed Ai and Bi that participate in the replication process.

The experimental results revealed an unexpected complication / quaternary structure variant in the replication process and the interpretations are interesting. Nevertheless, a few sentences / texts need to be corrected or amended to make the descriptions more precise and accurate.

In detail, the following items need to be addressed:

1) In the introduction (lines 86 & 87) the authors claim that "PMCA mimics in vivo prion replication with accelerated kinetics". Given the overall differences between in vivo prion conversion and PMCA-

driven prion misfolding / conversion, this statement is an over simplification. It would be useful to remind the reader about the principle differences and similarities between in vivo replication and ultrasound-driven in vitro conversion reactions.

2) On line 116 the term PrP "invalidated" mice is used instead of the more commonly used "Prnp-/- mice" or "knock-out PrP mice". It would best to use standard nomenclature.

3) In figures 1 A-C and S1 it is argued that small assemblies disappeared in favor of larger assemblies. Given the fact that the pre-symptomatic samples were concentrated to obtain a detectable signal level, it cannot be said that the oligomeric assemblies disappeared in favor of larger polymers. The relative proportions changed certainly, but the absolute amount of smaller assemblies may remain unchanged, since the larger aggregates dominate the Western blots.

4) On line 125 the authors write that "we used a high-throughput proprietary variant of PMCA (termed mb-PMCA)". If the method is not fully disclosed, then the results cannot be reproduced by others in the field. The method needs to be properly described.

5) On line 143 it is stated that "While a simple dilution of 127S seed drastically affected the size distribution of 127S PrPSc assemblies, sonication of concentrated 127S seeds in the PrP0/0 substrate revealed mostly the presence of large-sized assemblies." This behavior is unexpected and should be described in more detail to clarify this unexpected aggregation reversal.

6) On line 211/212 the authors write: "To further discard the existence of an equilibrium process...". The proper expression would be "to refute or disprove" a hypothesis.

Reviewer #3 (Remarks to the Author):

This is a very interesting and significant manuscript that suggests a new templating pathway for prion propagation, as alternative to the canonical templating model.

After decades of studying prion diseases, we still do not know for sure how prions work. In fact, the existence of many different models of prion structure during the last two decades shows the lack of knowledge about these intriguing pathogens. Here, it is the first time that a new secondary templating pathway is proposed.

Although the manuscript is very well written and the experiments carefully designed and performed, there are several parts that are not so easy to follow due to the complexity of the subject.

It would be very useful to have a cartoon showing with figures all the components of the new model. Ai, Bi, are reasonable understandable when are written but representative figures showing the different pathways in the figure 8 would be even better and it would be very helpful to support the mechanistic explanation.

I have only some comments and minors:

Line 120: The sentence "ii) the evolution of the pathology is concerted with a quaternary structural diversification of PrP assemblies" should be rephrased or explained better because it is not easy to understand.

Lines 143-147: "While a simple dilution...". These sentences should be explained better because the rationale behind them is difficult to follow.

Lines 170-171: The suggestion would be more understandable if a cartoon with figures is presented. It is especially difficult to visualize the concept "distinct polymerization pathways".

Line 176: "4 h" should be "3 h".

Line 183: The sentence "... strictly from a quaternary structure rearrangement through size increase rather than change in compactness reducing the hydrodynamic radius" should be rephrased or explained better.

Lines 189-191: It is not clear why the assemblies present in P2 enhance their own formation according to an autocatalytic process. This should be explained better because it is essential to understand the discussion. A practical example for "autocatalytic process" might help.

Lines 197-199: This sentence should be rephrased or explained better because it is not easy to understand.

Lines 223-224: This sentence should be rephrased or explained better because it is not easy to understand.

Line 329: The sizes for i_A and i_B are important. The way in which the authors conclude with the numbers "<5" and "around 20" should be elaborated.

Line 480: "tg20" should be "tga20".

Line 520: "5 M" should be "5 N".

Line 769: It would be useful to show the PrPres signals of the amplified samples by WB (for each dilution which the SV profile is shown).

Line 825: It should be explained that the figure is referring to 127S inoculum.

Line 870: I like the idea of "strain structural determinant (SSD)" but should be explained better.

Figure S3: "Intracranially" should be "intracerebrally".

Point by point response to reviewer's comments

Reviewer #1 (Remarks to the Author):

The authors provide a combination of empirical evidence and mechanistic modeling to provide novel insights into PrPSc assembly. The work is extensive, multifaceted, competently performed and quite thought-provoking. It should be of significant interest to the prion field and beyond.

Questions/Comments/suggestions:

1. There are a number of odd word choices that might be reconsidered:

L116: invalidated

Done. Replaced by knock-out.

L165, L189 and elsewhere: surface; do the authors mean area under the curve?

Sorry for the mistake lane 165, the graph in Figure 2B is referring to the percentage of assemblies at the peak maximum.

L189: Figure 2E is corresponding to the area under the curve.

The legends and figures have been modified accordingly to avoid any ambiguity.

L201 and elsewhere: kinetical \diamond kinetic

L395: favorized \diamond favored

Thank you, this has been corrected.

2. L129 & Fig 1: Need to show/describe SV analysis of PMCA seeded with normal brain.

3. L161: Similarly, what happens to P1 with a normal brain seed in mb-PMCA in this experiment?

We interpret by "PMCA seeded with normal brain" that the reviewer wants us to show the size distribution of PrP present in a PMCA product seeded with healthy brain homogenate as control.

We followed this recommendation and now provide this information as Supplementary Fig. 2 c-d, before and after PK treatment. As expected, PrP^C size distribution pattern remains unaffected by the PMCA conditions and no PrP^{res} can be detected. In particular, there is no spontaneous formation of P₁ (or P₂) PrP^{res}. The information is now mentioned in the results section, lanes 124-128.

4. L256: If B_i is bigger, why does it elute more slowly from SEC?

This is a very pertinent question. As we mentioned, SEC leads to separate the objects based on their hydrodynamic radius and *per se* based on their structural conformation. The sample preparation conditions for SEC (use of detergents and 200mM NaCl for ionic strength) disrupt PrP assemblies into their native elementary subunit (suPrP^A and suPrP^B). According to our SEC calibration and S200 column resolution, the low difference observed in the elution volume between suPrP^A and suPrP^B suggests a difference of hydrodynamic conformation rather than of molecular weight, suPrP^B being more compact than suPrP^A.

Therefore, even if suPrP^B has a more compact hydrodynamic radius, it has a propensity to form larger assemblies (B_i) due to its physicochemical properties.

To avoid any confusion regarding the size/conformation of suPrP^B and its propensity to form larger assemblies as B_i, we now mention in the revised version of the manuscript (Lanes 254-257 & 268-270)

that the structural difference between suPrP^A and suPrP^B confers them different physicochemical properties which dictate their aggregation propensity.

We would like to bring to the reviewer attention that the higher compactness of suPrP^B vs suPrP^A is in good accordance with the fact that it constitutes a thermodynamic-attractor, rendering the process of B_i formation quasi irreversible.

5. L 261, Fig 6: Why might the high-seed reaction products bifurcate into peaks that are elute both faster and slower than the low-seed products? These results, on the face of it, would suggest that smaller and larger particles are generated with high seed. How then might the smaller particles relate to suPrP^A? Might this be another elementary subunit?

This is a highly pertinent observation that did not escape our consideration and deep analysis. To discuss this point in further details, we provide as Figure 1 for the reviewer the chromatogram of the highest and lowest-seed concentration from Figure 6c and their respective peak decomposition.

Figure 1.

(A) Chromatograms of the 127S-PMCA products generated with the lowest (blue) and highest (red) seed concentrations, as reported in Figure 6C of the manuscript. The peak maxima are indicated by a dotted line. (B-C) Deconvolution of the low- (B) and high-seed (C) chromatograms using maximum entropy fitting method (dotted lines in B and C) with four gaussians. This deconvolution reveals that the relative surface area of the peak assigned to suPrP^B (pink curve with an elution volume of 15.5ml) increases with seed concentration. This increase is concerted with the apparition of a new assembly/population (orange curve with an elution volume of 14.1ml) which contributes to the left side-shift of the chromatogram. The green peak corresponds to an unassigned population, with an elution volume compatible with monomeric truncated PrP 27-30KDa. The red circles represent the amounts of PrP^{res}, as determined experimentally (see figure 6 in the manuscript).

As the referee can notice, PMCA performed with a high-seed concentration exhibits a double peak. The larger peak on the left-side of the chromatogram has shifted by almost three fractions towards higher molecular weight (i.e. fast eluted product) as compared to the major peak observed at low-seed concentration (Figure 1A for referee). It should be underlined that i) this shift is mostly restricted to the left side of the chromatogram as the right-side assigned to suPrP^B (according to figure 6A and 6B) remains unchanged, ii) this shift is not resulting from any experimental variability (see the error bars in figure 6A); due to the reproducibility of the SEC performed with a S200 column. Therefore, we agree with the referee that the shift toward higher Mw (fast eluted product) strongly suggests emergence of larger assemblies at high concentration seed that may be assigned to a new suPrP, as suggested by the referee.

There is, however, another trivial possibility we would like to mention. According to the kinetic scheme in figure 8, there are at least two equilibrium reactions between A_1 and B_1 and between B_1 and B_i . Thus, the progressive shift towards higher molecular weight of the chromatogram can be explained by an old and well-known phenomenon in SEC, as a consequence of an association and dissociation equilibrium displacement during the separation on the SEC column [1, 2]. Indeed, let's consider two objects A and B forming a complex AB according to the equilibrium $X + Y \rightleftharpoons XY$. If the dissociation constant (Kd) of this process is low enough (high affinity of X for Y), the SEC analysis will estimate the quasi-precise hydrodynamic radius (apparent Mw) of the complex XY. If the Kd is high enough, the chromatogram corresponding to the XY complex will elute at higher elution volume than the theoretical elution volume of the XY complex. Two elementary processes in the kinetic scheme proposed in equation 2, 3 and figure 8 from our manuscript could therefore explain the higher Mw product (i.e. faster product):

- 1- The B_i assemblies forming the P_2 peak in SV are in equilibrium with their $suPrP^B$ according to the equilibrium $B_i \rightleftharpoons B_{i-1} + suPrP^B$. The amount of $suPrP^B$ drives this equilibrium. One can imagine that high-seed concentration generates more $suPrP^B$. Therefore, this equilibrium will be shifted more toward $B_{i \geq 2}$ than $suPrP^B$. $B_{i \geq 2}$ having an apparent molecular weight higher than $suPrP^A$ and $suPrP^B$ taken independently. Thus, the fast product would be the chromatographic-echo of $B_2 \rightleftharpoons 2suPrP^B$.
- 2- The secondary templating pathway described in the present work requires the formation of a binary complex that we called 'C', which is a $suPrP^A:suPrP^B$ complex according to the equilibrium $suPrP^A + suPrP^B \rightleftharpoons C$. The higher amount of B_i (and *per se* of $suPrP^B$) generated at high-seed concentration could displace the equilibrium toward C ($suPrP^A:suPrP^B$ complex) with a higher molecular weight than $suPrP^A$ and $suPrP^B$ taken independently. Thus, the faster product would be a chromatographic-echo of the equilibrium $suPrP^A + suPrP^B \rightleftharpoons C$.

We intentionally did not detail this specific point because it is too preliminary. However, to address the reviewer remark, we added a comment on this point in the revised version of the results and in the discussion section, by indicating that the shift could result from the existence of association dissociation equilibrium (Lanes 263-265).

6. L296: Fig 3C: This fold increase seems large. My understanding is that this statement is based on the comparison of % P2 peak the PK- tg338 row with that of the PK- no-reconstitution row. The fold increase in the former is more like 1.2 fold. Perhaps I am misunderstanding what is meant here.

The referee observation is correct and we must apologize for this mistake. The precise fold increase is approx. 1.4. The text has been modified accordingly in the revised version (Lane 302).

7. L300-302: The authors state: "The appearance of a low amount of B_i after a long incubation period without PrPC may result from the leakage of monomers from a conformer cosedimenting with A_i ." Might this have something to do with the smaller peaks in Fig 6A and C?

This is an interesting hypothesis that we also considered in our analysis. This hypothesis is particularly relevant as the position of the aforementioned smaller peaks corresponds to truncated, monomeric PrP (PrP 27-30kDa) that could result from a slow depolymerization into PrP (see Figure 1 for referee). The depolymerization into monomeric PrP is interesting for two main reasons: i) the origin of this leakage could be another PrP^{res} subset thermodynamically less stable which would depolymerize into monomer in the experimental conditions; ii) the existence of a monomeric PrP exchange between A and B assemblies.

We intentionally did not expand these hypotheses, which obviously would need further experimental investigation and are at some point out of the main topic of our present work.

Reviewer #2 (Remarks to the Author):

The manuscript entitled “Quaternary structural convergence and structural diversification of prion assemblies at the early replication stage” (COMMSBIO-19-0516-T) by Igel-Egalon et al. describes quaternary structural conversions in in vivo and in vitro models of prion replication.

The misfolding of the prion protein from the cellular PrP^c to the disease-associated PrP^{Sc} is thought to progress through oligomeric intermediate states. The dynamic and temporary nature of protein oligomers makes them a difficult research subject in general. In this manuscript the authors describe a technically and conceptually original approach to investigate the aggregation state and conformation of prion protein oligomers during in vivo and in vitro conversions. By using sedimentation velocity ultracentrifugation approaches the authors discovered two distinct oligomeric populations termed A_i and B_i that participate in the replication process.

The experimental results revealed an unexpected complication / quaternary structure variant in the replication process and the interpretations are interesting. Nevertheless, a few sentences / texts need to be corrected or amended to make the descriptions more precise and accurate.

In detail, the following items need to be addressed:

1) In the introduction (lines 86 & 87) the authors claim that “PMCA mimics in vivo prion replication with accelerated kinetics”. Given the overall differences between in vivo prion conversion and PMCA-driven prion misfolding / conversion, this statement is an over simplification. It would be useful to remind the reader about the principle differences and similarities between in vivo replication and ultrasound-driven in vitro conversion reactions.

It was meant there: PMCA mimics in vivo replication with accelerated kinetics, with respect to the amount of infectivity generated in the brain. To save space, the sentence has been moved lanes 115-188 and the *ad hoc* references have been added.

2) On line 116 the term PrP “invalidated” mice is used instead of the more commonly used “Prnp^{0/0}-mice” or “knock-out PrP mice”. It would best to use standard nomenclature.

This has been corrected, thank you.

3) In figures 1 A-C and S1 it is argued that small assemblies disappeared in favor of larger assemblies. Given the fact that the pre-symptomatic samples were concentrated to obtain a detectable signal level, it cannot be said that the oligomeric assemblies disappeared in favor of larger polymers. The relative proportions changed certainly, but the absolute amount of smaller assemblies may remain unchanged, since the larger aggregates dominate the Western blots.

We agree that in the absence of internal reference for PrP quantification, the present statement is not exact. The revised version of the manuscript has been modified accordingly. The sentence L118 “small assemblies mostly disappeared at the expense of larger assemblies” was replaced by: “larger assemblies were observed” (Lanes 109-110).

4) On line 125 the authors write that “we used a high-throughput proprietary variant of PMCA (termed mb-PMCA)”. If the method is not fully disclosed, then the results cannot be reproduced by others in the field. The method needs to be properly described.

Sorry for the misuse of the adjective “proprietary”. The method has been previously described in details. We realized that we omitted to introduce the mb-PMCA references in the results section, even if in the Material and Methods section the mb-PMCA method is described and the seminal publications

cited as reference. The sentence lanes 115-118 has been rephrased accordingly and the term proprietary removed.

5) On line 143 it is stated that “While a simple dilution of 127S seed drastically affected the size distribution of 127S PrP^{Sc} assemblies, sonication of concentrated 127S seeds in the PrP⁰/0 substrate revealed mostly the presence of large-sized assemblies.” This behavior is unexpected and should be described in more detail to clarify this unexpected aggregation reversal.

The effect of dilution on the size distribution of PrP^{Sc} assemblies has been detailed in our previous work in 2017: we showed by static light scattering approaches that PrP^{Sc} assemblies (from hamster 263K prions) are in a dynamic equilibrium with their oligomeric elementary building block (suPrP) and that a simple dilution induces PrP^{Sc} depolymerization and structural rearrangement towards suPrP [3, 4]. In contrast, in the present manuscript, the sonication conditions used for PMCA amplification did not induce the formation of small PrP^{Sc} assemblies, even though fragmentation is commonly admitted to be at the base of the PMCA amplification process.

We agree with the reviewer that the weight of this sentence is important in general and requires more explanation. We added reference to our previous work and a recent review on this topic, that describes reversal aggregation (depolymerization) and highlights the dynamic aspect of the quaternary structure of PrP^{Sc} assemblies commonly admitted to be deadpan. We have substantially modified the text to highlight these points in the revised version of the manuscript (lanes 135-139).

6) On line 211/212 the authors write: “To further discard the existence of an equilibrium process...”. The proper expression would be “to refute or disprove” a hypothesis.

This has been corrected accordingly.

Reviewer #3 (Remarks to the Author):

This is a very interesting and significant manuscript that suggests a new templating pathway for prion propagation, as alternative to the canonical templating model. After decades of studying prion diseases, we still do not know for sure how prions work. In fact, the existence of many different models of prion structure during the last two decades shows the lack of knowledge about these intriguing pathogens. Here, it is the first time that a new secondary templating pathway is proposed.

Although the manuscript is very well written and the experiments carefully designed and performed, there are several parts that are not so easy to follow due to the complexity of the subject.

It would be very useful to have a cartoon showing with figures all the components of the new model. A_i , B_i , are reasonable understandable when are written but representative figures showing the different pathways in the figure 8 would be even better and it would be very helpful to support the mechanistic explanation.

We totally agree with the reviewer that a cartoon at some point could be useful. However, the kinetic process (as an ensemble) that we describe here involves several equilibria between A_i and B_i and their elementary subunit. A cartoon describing all these processes should also describe the size of these elementary subunits, which according to the present study is still uncertain. The number of PrPC involved to generate suPrP^B is not estimable through our approaches. Therefore, to avoid building a cartoon that may sound too speculative, we would like to stay “rude” and maintain a “chemical”-like kinetic scheme.

1) Line 120: The sentence “ii) the evolution of the pathology is concerted with a quaternary structural diversification of PrP assemblies” should be rephrased or explained better because it is not easy to understand.

The whole conclusion has been rephrased to facilitate its understanding (lanes 110-113).

“...These observations suggest that, in the brain, the replication process follows two phases that are common to the three prion strains: i) a first phase generating mainly small oligomers equivalent in size (at the SV resolution); ii) a second phase of quaternary structural diversification during the disease evolution.”

2) Lines 143-147: “While a simple dilution...”. These sentences should be explained better because the rationale behind them is difficult to follow.

See response to reviewer #2, point 5).

3) Lines 170-171: The suggestion would be more understandable if a cartoon with figures is presented. It is especially difficult to visualize the concept “distinct polymerization pathways”.

We modified the text in order to better explain the concept of “distinct polymerization pathway” (lanes 161-169). However, at this stage of the manuscript and due to the fact that we don’t have any information about the active phase of PMCA and how the first B_i assemblies are generated, a cartoon will be too speculative.

4) Line 176: “4 h” should be “3 h”.

This has been corrected, thank you.

5) Line 183: The sentence “... strictly from a quaternary structure rearrangement through size increase rather than change in compactness reducing the hydrodynamic radius” should be rephrased or explained better.

Thank you for pointing that. We simplified this sentence, which was not clear (lanes 179-182).

“...This observation implies that the low sedimentation velocity of the assemblies forming P_1 does not result from an interaction with lipids or other low-density molecules and that the sedimentation velocity increase of P_2 compared to P_1 results strictly from a quaternary structure rearrangement through size increase.”

6) Lines 189-191: It is not clear why the assemblies present in P_2 enhance their own formation according to an autocatalytic process. This should be explained better because it is essential to understand the discussion. A practical example for “autocatalytic process” might help.

We have rephrased the sentence accordingly and provide a general reference (lanes 187-190).

“...Analyzing the time-dependent surface variation of P_1 and P_2 showed that the formation of P_2 assemblies started slowly, increased steadily from ~2-3h up to ~7h and finally reached a plateau. This sigmoidal variation is hallmark of an autocatalytic reaction[5] and indicates that the assemblies present in P_2 enhance their own formation.”

7) Lines 197-199: This sentence should be rephrased or explained better because it is not easy to understand.

The sentence has been further explained (lanes 196-201). The initial sentence “...this size translation contrasts with the bimodal phase (transformation of P_1 to P_2) and highlights a change in the kinetic regime compatible with a coalescence process...” has been replaced by “...This size translation deeply contrasts with the bimodal phase (transformation of P_1 to P_2) observed during the 0 to 7-day quiescent incubation and highlights a change in the kinetic regime. This new regime would be compatible with a coalescence process (references), whereby assemblies would grow by end-to-end or lateral association rather than by incorporation of monomers (Supplementary Fig. 5).” We also refer to Supplementary Fig 5. which shows a mathematical simulation of size distribution evolution according to a coalescence process.

8) Lines 223-224: This sentence should be rephrased or explained better because it is not easy to understand.

The original sentence “...The irreversible transformation of P_1 to P_2 argues in favour of the existence of two sets of PrPres assemblies forming P_1 and P_2 peaks. Only a thermodynamically “locking” process i.e. a structural rearrangement and formation of higher stable object can be at the origin of such irreversibility” has been simplified and replaced by “...The irreversible character of the transformation of P_1 to P_2 argues in favour of the existence of a thermodynamically-driven “locking” process. This implies structural rearrangements of P_1 assemblies and formation of higher stable P_2 objects. “ (lanes 221-225).

9) Line 329: The sizes for i_A and i_B are important. The way in which the authors conclude with the numbers “<5” and “around 20” should be elaborated.

As stated lane 328 of the original manuscript, the size expressed in protomer number has been estimated according to our SV calibration with “PrP oligomers and globular mass markers”. This information and details are referenced in the manuscript by Tixador et al. [6]. This reference is also included in lane 328.

10) Line 480: “tg20” should be “tga20”.

11) Line 520: “5 M” should be “5 N”.

Corrected, thank you.

12) Line 769: It would be useful to show the PrP^{res} signals of the amplified samples by WB (for each dilution which the SV profile is shown).

The PrP^{res} signals of the amplified samples are now shown as Supplementary Fig. 4. The amounts are similar, whatever the seed dilution, as previously observed [7, 8]. The reference to this figure is now lanes 159-160.

13) Line 825: It should be explained that the figure is referring to 127S inoculum.

Done, thank you.

14) Line 870: I like the idea of “strain structural determinant (SSD)” but should be explained better.

The strain structural determinant corresponds to PrP domain(s) harboring the replicative and strain information. This idea has been conceptualized in our previous publications [3, 4]. The sentence has been modified accordingly.

15) Figure S3: “Intracranially” should be “intracerebrally”.

Corrected, thank you.

References

1. Bao, J.; Krylova, S. M.; Cherney, L. T.; Le Blanc, J. C.; Pribil, P.; Johnson, P. E.; Wilson, D. J.; Krylov, S. N., Pre-equilibration kinetic size-exclusion chromatography with mass spectrometry detection (peKSEC-MS) for label-free solution-based kinetic analysis of protein-small molecule interactions. *Analyst* **2015**, *140*, (4), 990-4.
2. Jonker, N.; Kool, J.; Irth, H.; Niessen, W. M., Recent developments in protein-ligand affinity mass spectrometry. *Anal Bioanal Chem* **2011**, *399*, (8), 2669-81.
3. Igel-Egalon, A.; Bohl, J.; Moudjou, M.; Herzog, L.; Reine, F.; Rezaei, H.; Beringue, V., Heterogeneity and Architecture of Pathological Prion Protein Assemblies: Time to Revisit the Molecular Basis of the Prion Replication Process? *Viruses* **2019**, *11*, (5).
4. Igel-Egalon, A.; Moudjou, M.; Martin, D.; Busley, A.; Knapple, T.; Herzog, L.; Reine, F.; Lepejova, N.; Richard, C. A.; Beringue, V.; Rezaei, H., Reversible unfolding of infectious prion assemblies reveals the existence of an oligomeric elementary brick. *PLoS Pathog* **2017**, *13*, (9), e1006557.
5. Moore, J. W.; Pearson, R. G., *Kinetics and Mechanism*. 1981; p 480.
6. Tixador, P.; Herzog, L.; Reine, F.; Jaumain, E.; Chapuis, J.; Le Dur, A.; Laude, H.; Beringue, V., The physical relationship between infectivity and prion protein aggregates is strain-dependent. *PLoS Pathog* **2010**, *6*, (4), e1000859.
7. Moudjou, M.; Chapuis, J.; Mekrouti, M.; Reine, F.; Herzog, L.; Sibille, P.; Laude, H.; Vilette, D.; Andreoletti, O.; Rezaei, H.; Dron, M.; Beringue, V., Glycoform-independent prion conversion by highly efficient, cell-based, protein misfolding cyclic amplification. *Sci Rep* **2016**, *6*, 29116.
8. Moudjou, M.; Sibille, P.; Fichet, G.; Reine, F.; Chapuis, J.; Herzog, L.; Jaumain, E.; Laferrriere, F.; Richard, C. A.; Laude, H.; Andreoletti, O.; Rezaei, H.; Beringue, V., Highly infectious prions generated by a single round of microplate-based protein misfolding cyclic amplification. *MBio* **2014**, *5*, (1), e00829-13.

REVIEWERS' COMMENTS:

Reviewer #1 (Remarks to the Author):

The authors have addressed my concerns adequately.

Reviewer #2 (Remarks to the Author):

The revised manuscript entitled "Quaternary structural convergence and structural diversification of prion assemblies at the early replication stage" (COMMSBIO-19-0516A) by Igel-Egalon et al. is substantially improved over the original submission.

Some of the changes that were made during the revision process improved the readability and clarified issues that were insufficiently explained before. Interestingly, some additions opened up new questions, but these can be left for another occasion / further study.